# TRAP1 chaperone protein mutations and autoinflammation

Ariane SI Standing[1,2] , Ying Hong[1], Coro Paisan-Ruiz[3] , Ebun Omoyinmi[1], Alan Medlar[4], Horia Stanescu[4], Robert Kleta[4], Dorota Rowcenzio[5], Philip Hawkins[5], Helen Lachmann[5], Michael F McDermott[6] , Despina Eleftheriou[1], Nigel Klein[1], Paul A Brogan[1]

We identified a consanguineous kindred, of three affected children with severe autoinflammation, resulting in the death of one sibling and allogeneic stem cell transplantation in the other two. All three were homozygous for *MEFV* p.S208C mutation; however, their phenotype was more severe than previously reported, prompting consideration of an oligogenic autoinflammation model. Further genetic studies revealed homozygous mutations in *TRAP1*, encoding the mitochondrial/ER resident chaperone protein tumour necrosis factor receptor associated protein 1 (TRAP1). Identification of a fourth, unrelated patient with autoinflammation and compound heterozygous mutation of *TRAP1* alone facilitated further functional studies, confirming the importance of this protein as a chaperone of misfolded proteins with loss of function, which may contribute to autoinflammation. Impaired TRAP1 function leads to cellular stress and elevated levels of serum IL-18. This study emphasizes the importance of considering digenic or oligogenic models of disease in particularly severe phenotypes and suggests that autoinflammatory disease might be enhanced by bi-allelic mutations in *TRAP1*.

## Introduction

Many cellular functions generate by-products which are tightly managed to maintain cellular homeostasis. When these homeostatic pathways are perturbed, cellular stress ensues (Galluzzi et al, 2018). Cellular stress can be physiologically vital for host survival. For example, immune cellular stress in response to pathogen-related signals leads to the activation of important immune pathways that are vital to resolve infection and promote tissue repair. It is now recognized that the innate immune system has evolved to modulate the negative effects of pathogens on host homeostasis (Soares et al, 2017). For example, the NLRP3 inflammasome can recognize damage-associated molecular patterns and other signs of cellular damage, including reactive oxygen species (ROS), and it responds to these rapidly by assembling its components to release of IL-1 and IL-18, important pro-inflammatory cytokines central to many innate immune responses (van de Veerdonk et al, 2011; Zhou et al, 2011). Dysregulation of innate immune cellular homeostasis can result from genetic mutation, causing a growing number of monogenic autoinflammatory diseases resulting from dysregulation of key inflammatory pathways (Manthiram et al, 2017). One such pathway is the unfolded protein response which responds to accumulation of intracellular misfolded proteins, important in the pathogenesis of several autoinflammatory diseases (Park et al, 2012). In particular, patients with tumour necrosis factor receptor associated periodic syndrome (TRAPS) have a heterozygous mutation of *TNFRSF1A* which causes TNFR1 to fold incorrectly, activating the unfolded protein response pathway in the ER causing increased intracellular ROS, and dysregulated production of IL-1β and other pro-inflammatory cytokines resulting in autoinflammation (Simon et al, 2010; Bulua et al, 2011; Dickie et al, 2012). Another example is the buildup of damaged, oxidised proteins driving the pathogenesis of the proteasome disability syndromes (Agarwal et al, 2010; Arima et al, 2011; Kitamura et al, 2011; Liu et al, 2011; Kanazawa, 2012). More recently, mutations in *TRNT1* encoding a nucleotidyltransferase required for the maturation of cytosolic and mitochondrial tRNAs, essential for protein synthesis, has been linked to congenital sideroblastic anemia with immunodeficiency, fevers, and developmental delay (SIFD). Loss of TRNT1 activity disrupts protein homeostasis leading to mitochondrial damage, ROS production and autoinflammation (Wiseman et al, 2013; Chakraborty et al, 2014; Giannelou et al, 2018).

In this study, we have identified a consanguineous kindred, with three affected children with a very severe autoinflammatory disease, resulting in the death of one sibling and need for allogeneic haematopoietic stem cell transplantation (allo-HSCT) in the two surviving siblings. They harboured the recently described pathogenic homozygous p.S208C mutation in *MEFV*. This gene is mutated in patients with familial Mediterranean fever (FMF) and pyrin-associated

[1]University College London and Great Ormond Street Institute of Child Health, London, UK   [2]The Natural History Museum, London, UK   [3]Department of Neurology, Icahn School of Medicine at Mount Sinai, New York City, NY, USA   [4]University College London Division of Medicine, London, UK   [5]National Amyloidosis Centre and Royal Free Hospital, London, UK   [6]Leeds Institute of Rheumatic and Musculoskeletal Medicine, St James's University Hospital, Leeds, UK

Correspondence: ariane.standing@ucl.ac.uk

autoinflammation with neutrophilic dermatosis (PAAND) (Masters et al, 2016; Moghaddas et al, 2017). We recently reported homozygous *MEFV* p.S208C and p.S208T mutation as the cause of autoinflammation with recurrent fevers, erythematous skin rashes, peripheral blood eosinophilia, and intestinal inflammation (Hong, 2018). However, the phenotype in the kindred studied herein was much more severe than that associated with our previous report of autoinflammation caused by homozygous *MEFV* p.S208 mutation. Therefore, further genetic studies were undertaken, ultimately revealing a homozygous variant in a gene in the same genomic region, *TRAP1*, which encodes the mitochondrial chaperone protein tumour necrosis factor receptor associated protein 1 (TRAP1). Identification of a fourth, unrelated patient with autoinflammation in association with compound heterozygous variants in *TRAP1* (without mutation in any other known autoinflammatory genes) suggested that mutant TRAP1 may contribute to autoinflammation, via increased ROS generation. These findings emphasize the importance of considering digenic or oligogenic models of disease in patients with particularly severe autoinflammatory phenotypes.

## Results and Discussion

Three siblings from a consanguineous (first-cousin) Pakistani family presented with a severe autoinflammatory phenotype. The index case (IV-1, Fig 1A–D) first presented at 2 wk of age with severe oral and lip ulceration, nasal inflammation (Fig 1A–C), recurrent fevers without obvious periodicity or evidence of infection every 4–6 wk and lasting several days, nasal septal collapse due to chronic inflammation (Fig 1A), a transient maculopapular rash (Fig 1D), and failure to thrive. There was an intense and continuous acute phase response, with persistently elevated serum amyloid A protein (SAA) and C reactive protein (CRP) concentrations usually exceeding 100 mg/l (reference range < 10 for both; Fig 1I). Levels of serum IgD persistently exceeded the upper limit of detection of 900 IU/ml for the assay (reference range < 100). The patient did not have mevalonic aciduria, and routinely available genetic screening for the common causes of autoinflammatory disease revealed wild-type for *MVK, TNFRSF1A, NLRP3*, and *MEFV* (exons 10 and part of exon 2, which at the time was offered in routine clinical care). Further details of all the immunological and histological investigations undertaken are provided in Table S1. The original best fit clinical description made more than 20 yr ago was that of "congenital Wegener's granulomatosis" (Hoffman et al, 1992), although anti-neutrophil cytoplasmic antibodies against proteinase 3 or myeloperoxidase were repeatedly negative. Various therapies (all at standard paediatric doses, Table S1) were tried including, non-steroidal anti-inflammatory drugs, daily prednisolone (0.3 to 2 mg/kg/d), pulsed intravenous methylprednisolone, colchicine, cyclophosphamide, azathioprine, mycophenolate mofetil, thalidomide, rituximab and anakinra, all with limited or no success. She received a single dose of infliximab (a murine-human chimeric monoclonal antibody against tumour necrosis factor); however, the patient deteriorated with worsening of rash and fever in the first 3 d after the infusion; infliximab was subsequently discontinued. Her symptoms were partially controlled with high-dose daily prednisolone, although there continued to be an intense persistent

acute phase response (Fig 1I). In view of ongoing chronic inflammation with chronically elevated SAA and CRP, corticosteroid dependency, and failure to thrive, she underwent allogeneic HSCT from a fully matched (10/10) nonrelated donor at the age of 13 yr. Within 1 mo, there was complete resolution of all her symptoms, normalisation of her inflammatory markers and IgD levels, and over time, normalisation of growth. She engrafted fully (100% donor chimerism) with no major complications and remains well and off all medication now 12 yr posttransplantation. Her younger female sibling (IV-2) also had similar symptoms, persistently elevated acute-phase reactants (Fig 1I), and high serum IgD (Table S1). Her symptoms and serological markers were initially partially controlled by anakinra 2–3 mg/kg/d; this clinical and serological response was short-lived, however (Fig 1I). Thus, she also underwent allogeneic HSCT from a fully matched nonrelated donor, which also cured her illness and she remains well and off all medication now 10 yr posttransplantation. A third sister (IV-4) presented at 2 mo of age with a severe clinical phenotype similar to the index case but with the addition of multifocal sterile osteomyelitis with lytic bony lesions and neutrophil infiltration on bone biopsy (Fig 1E–G and Table S1), including the cervical spine that required surgical stabilisation. This infant died at age 12 mo from acute cervical myelopathy from cervical vertebral collapse. The parents and two unaffected siblings were also formally assessed for the possibility of autoinflammation. Detailed past medical history and examination did not indicate any clinical phenotype. In addition, we measured SAA and CRP in the parents and both unaffected siblings and did not find any indication of subclinical inflammation. Further details of investigations and therapies received by all three patients are provided in Table S1.

Homozygosity mapping and parametric multipoint linkage analysis of the three affected children, two unaffected siblings, and both parents in this pedigree identified a single 5-Mb region on chromosome 16 that segregated with the disease within the family. Targeted resequencing of this region identified 118 coding change variants in 53 genes. Because of the unusual clinical presentation, we eliminated variants present with a frequency of more than 0.01 in the 1,000 genomes database (http://www.1000genomes.org), the Exome Variant Server (ESP) (https://evs.gs.washington.edu/EVS), the Exome Aggregation Consortium database (now aggregated into GnomAD), the Genome Aggregation Database (GnomAD) database (https://gnomad.broadinstitute.org/), and the Greater Middle East database (http://igm.ucsd.edu/gme/). After screening 100 healthy controls from the same Pakistani tribe (exact tribe withheld to maintain confidentiality), with further elimination of unlikely candidates, and further screening of the extended kindred (Fig 1H), two variants of interest remained. The first was a nonsynonymous homozygous variant in a previously unscreened region of exon 2 of *MEFV* c.622A>T, causing a p.S208C amino acid substitution. This variant was of considerable interest because mutations in *MEFV* are associated with the autoinflammatory diseases, FMF and PAAND (Masters et al, 2016; Moghaddas et al, 2017). We recently reported that homozygous MEFV substitution of serine 208 with threonine or cysteine causes an autoinflammatory disease associated with recurrent fevers, skin rashes, intestinal inflammation, and peripheral blood eosinophilia (Hong, 2018). This MEFV amino acid residue is part of the 14.3.3 binding motif; mutation in another key residue of

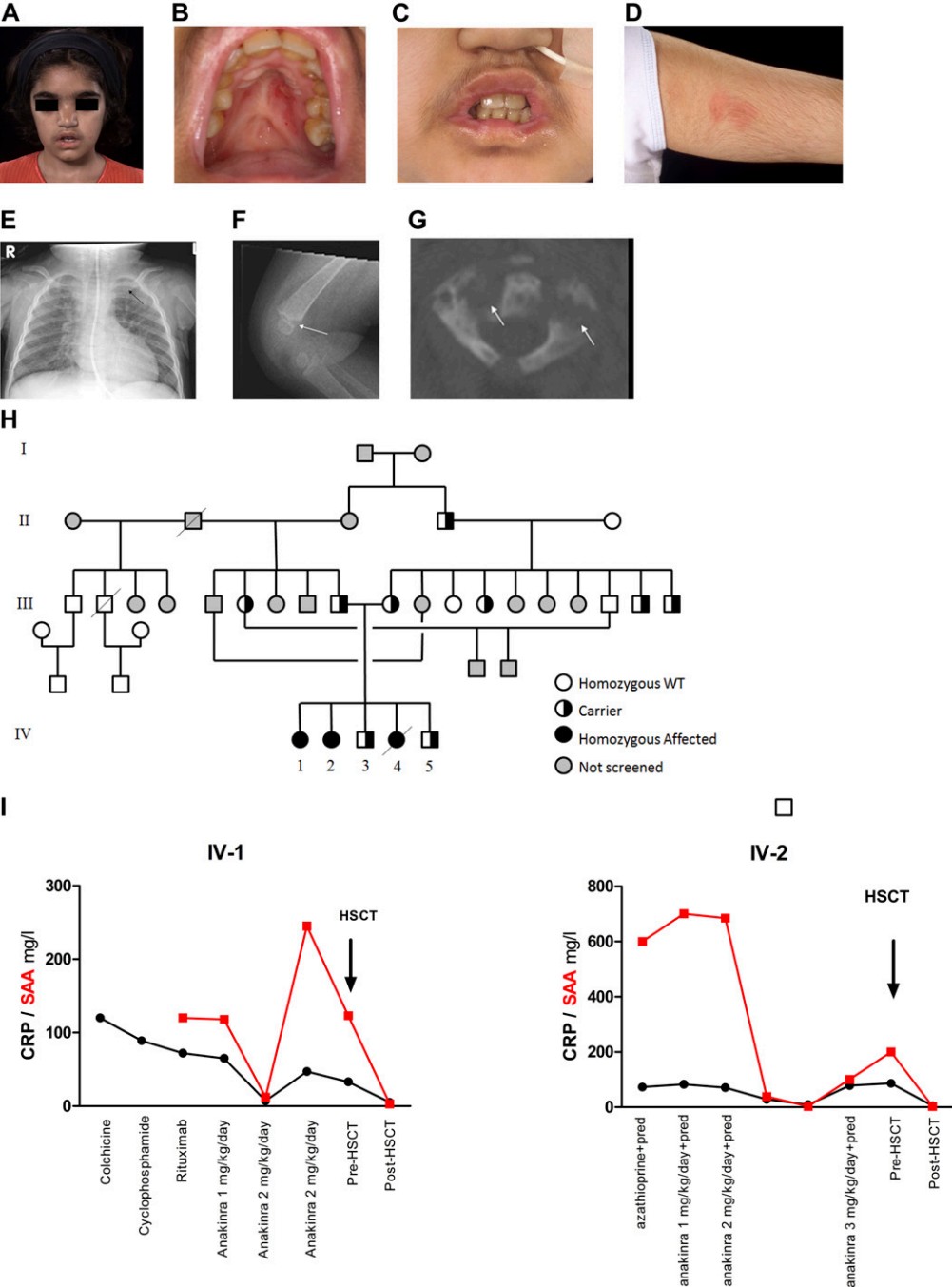

**Figure 1. Clinical phenotype and response to treatment.**
**(A)** Recurrent nasal inflammation resulting in cartilaginous nasal septal collapse and "saddle nose deformity" in the index case IV-1. **(B)** chronic ulceration of the hard palate leading to scarring (IV-1). Biopsy of mucosal ulcers revealed neutrophilic infiltration (see Table S1 for full histological descriptions). **(C)** ulcers also affected the lips causing scarring (IV-1). Note also the nasogastric feeding tube required for nutritional support for failure to thrive. **(D)** erythematous rash on upper arm (IV-1). **(E, F, G)** Sterile multifocal osteomyelitis in Patient IV-4, resulting in expansion and lytic inflammation in the medial end of the left clavicle (E) (arrowed), and of (F) distal femur (arrowed), and multiple cervical vertebra, as depicted in (G) showing a computer tomography scan of a cervical vertebra with lytic lesions (arrowed). Cervical myelopathy from vertebral collapse was the cause of death in Patient IV-4. **(H)** Family pedigree showing status of *TRAP1* mutation and the linked *MEFV* pathological variants. **(I)** Acute phase reactants plotted against treatments in IV-1, and IV-2, black line denotes C reactive protein, red lines serum amyloid A. WT, wild-type; HSCT, haematopoietic stem cell transplantation.

this motif (S242) is associated with the dominant autoinflammatory disease PAAND (Masters et al, 2016; Moghaddas et al, 2017). The particularly severe phenotype of the kindred studied herein prompted consideration of other possibly contributory genetic mutations in a digenic or oligogenic disease model. We identified a second homozygous variant of interest in close proximity (<0.5 Mb) to this *MEFV* variant, a nonsynonymous variant in the gene for the mitochondrial protein tumour necrosis factor receptor associated protein 1 (*TRAP1*) c.383G>A, leading to a p.R128H amino acid change, classified as damaging by three in silico predictive tools (Polyphen2

[Adzhubei et al, 2010], SIFT [Kumar et al, 2009], and MutationTaster [Schwarz et al, 2010]). TRAP1 is a member of the heat shock protein 90 family, a collection of chaperone proteins involved in the folding of nascent and damaged proteins that assist in the stabilisation of these proteins under cellular stress (Chen et al, 2005). TRAP1 is involved in managing unfolded proteins in the mitochondria and the ER (Chien et al, 2011; Matassa et al, 2011; Takemoto et al, 2011). This function reduces mitochondrial damage and, thus, the production of ROS. ROS are important activators of innate immunity, and elevated levels of mitochondrial ROS (mROS) have been

implicated in the pathogenesis of a number of autoinflammatory diseases (Tassi et al, 2010; Borghini et al, 2011; Bulua et al, 2011; Carta et al, 2012; Dickie et al, 2012; Omenetti et al, 2013; Giannelou et al, 2018); thus, impairment of functional TRAP1 protein might exacerbate any pre-existing autoinflammatory propensity.

We subsequently included *TRAP1* in a targeted next-generation sequencing panel used for routine diagnostic purposes (Omoyinmi et al, 2017), and in that context, screened this gene in 315 patients with suspected autoinflammation. This led to the identification of a fourth, unrelated patient (patient 4) with variants in this gene. This was a Caucasian adult female (of adopted parents, hence no biological family history available) with an autoinflammatory disease characterised by episodic fevers and systemic inflammation from early in life, cervical lymphadenopathy, arthralgia, elevated acute-phase reactants, including high SAA, partial response to corticosteroids, and complete response to anakinra during these inflammatory exacerbations. In total, she had three rare variants in *TRAP1*, all predicted to be damaging: these included the same c.383 G>A (p.R128H) variant (heterozygous) as identified in our kindred and two other heterozygous variants, c.947 G>A (p.R316H), and c.1330T>A (p.Y444N). To confirm variant segregation, we used TA cloning in this individual (TOPO-TA kit; Invitrogen) and demonstrated that p.R316H and p.Y444N segregated together, with p.R128H segregating on the other chromosome. This patient had no other mutations identified in any other autoinflammatory gene (including *MEFV*) included in our targeted panel (Omoyinmi et al, 2017). Subsequently, three *TRAP1* p.R128H homozygotes have been identified in the Exome Aggregation Consortium database, and six in the gnomAD database. The frequency of this allele in South Asian populations is 0.01 in the gnomAD database. Penetrance of this variant may be incomplete, and we suspect that it may enhance the inflammatory phenotype in combination with other mutations such as the *MEFV* p.S208C mutations or the additional *TRAP1* variants within this study, and may be less severe in isolation.

TRAP1 has a role in both the ER and the mitochondria, in particular refolding proteins and thus reducing mROS levels, ER stress, and cellular damage (Matassa et al, 2011). Given the role that ROS is known to play in inflammation (Zhou et al, 2003, 2011; Naik & Dixit, 2011; Mittal et al, 2014) and specifically in autoinflammation (Kirkali et al, 2008; Tassi et al, 2010; Bulua et al, 2011; Carta et al, 2012; Dickie et al, 2012; Omenetti et al, 2013; Giannelou et al, 2018), we measured the levels of oxidative stress and mROS in PBMCs from patient 4 and in three healthy age-matched controls using flow cytometry. PBMCs were not available for these experiments from any of the children in our pedigree (Fig 1H) because two had already undergone allogeneic HSCT and the third was deceased. MitoSox (MS) fluorescence is proportional to superoxide levels in the mitochondria, and H$_2$DCFDA fluorescence measures the accumulation of peroxylated proteins and overall intracellular oxidative stress. Compared with healthy controls, mROS were significantly elevated in the lymphocytes of patient 4 ($P < 0.001$), but not in monocytes (Fig 2A and B). Overall intracellular oxidative stress was elevated in both lymphocytes ($P < 0.001$) and monocytes ($P < 0.001$) of patient 4, compared with healthy controls (Fig 2C and D), suggesting that the compound heterozygous variants in *TRAP1* alone increase cellular oxidative stress and might drive autoinflammation in this patient. One limitation of our results, however, is that we did not look at specific T lymphocyte subsets. This is of potential importance becuase altered

mtROS production in T lymphocytes might also be a consequence rather than a cause of the autoinflammatory syndrome (Rashida Gnanaprakasam et al, 2018).

In the absence of PBMCs from the affected siblings in our informative family, we explored whether defective TRAP1 or pyrin was the predominant driver of cellular oxidative stress by assessing ROS levels in THP1 cells differentiated with PMA overnight to a macrophage-like phenotype, and then transfected with siRNA targeted against *MEFV* or *TRAP1*. On average, these knockdown (kd) siRNA transfection experiments reduced protein levels for both gene products by 50%, 48 h posttransfection (Fig S1). Superoxide (O$_2^-$) is highly labile and does not diffuse far before reacting with the cellular components and generating other free radicals. *TRAP1* kd led to increased levels of superoxide within the mitochondria as measured by MS fluorescence, although not in the cytoplasm (Fig 2E and F) compared with scrambled siRNA control cells. *MEFV* siRNA-mediated silencing of MEFV resulted in a small increase in cytoplasmic superoxide, as measured by DHE fluorescence and had no significant effect on mitochondrial superoxide levels (Fig 2E and F). ROS levels have been shown to be increased in FMF patients with highly penetrant mutations; thus, our finding of slightly increased cytoplasmic ROS in these *MEFV* kd experiments could indicate that cytoplasmic sources such as NADPH oxidases may be important in FMF (Omenetti et al, 2013). When the rates of O$_2^-$ production were compared using electron spin resonance, this was markedly increased in the TRAP1 kd cells but not in the MEFV kd (Fig 2G). This rate of O$_2^-$ production was sufficient to induce significant cellular oxidative stress, detected as an increase in DCF fluorescence after TRAP1 kd, but not MEFV kd (Fig 2H).

Recessive mutations in *TRAP1* have been associated with congenital abnormalities of the kidney and urinary tract (CAKUT), and with vertebral defects, anal atresia, cardiac defects, tracheoesophageal fistula, renal anomalies, and limb abnormalities (VACTERL) in five families (Saisawat et al, 2014). These malformation syndromes are not associated with an autoinflammatory phenotype. Most of these mutations were in the HSP90 domain of the protein, with each patient in that report harbouring at least one variant at this location; only one patient had a heterozygous variant affecting the ATP binding domain where the p.R128H *TRAP1* variant lies and no patient was homozygous or compound heterozygous for variants in this domain. Sequence analysis places the p.R128H variant within the first α helix of the ATP-binding Bergerat fold in the highly conserved N-terminal domain (Tanaka et al, 1998; Dutta & Inouye, 2000; Chen et al, 2005). The two additional rare (and predicted damaging) *TRAP1* variants found in patient 4 also affect α helices and ATPase activity (Table S2). The *TRAP1* p.R128H variant was present in all four of our patients: it was homozygous in the affected individuals in our informative pedigree and compound heterozygous with the two other variants in the unrelated patient 4. *TRAP1* p.R128H is predicted to be damaging by three different in silico prediction software packages (Polyphen2; SIFT; and MutationTaster) (Kumar et al, 2009; Adzhubei et al, 2010; Schwarz et al, 2010). Arginine at position 128 of TRAP1 is conserved throughout the bilateria, when this protein originated. The N-terminal domain has the highest sequence conservation across all HSP90 paralogues, and in this broader family, arginine-128 is conserved throughout the Eukaryota. To determine specifically if the *TRAP1* p.R128H variant could

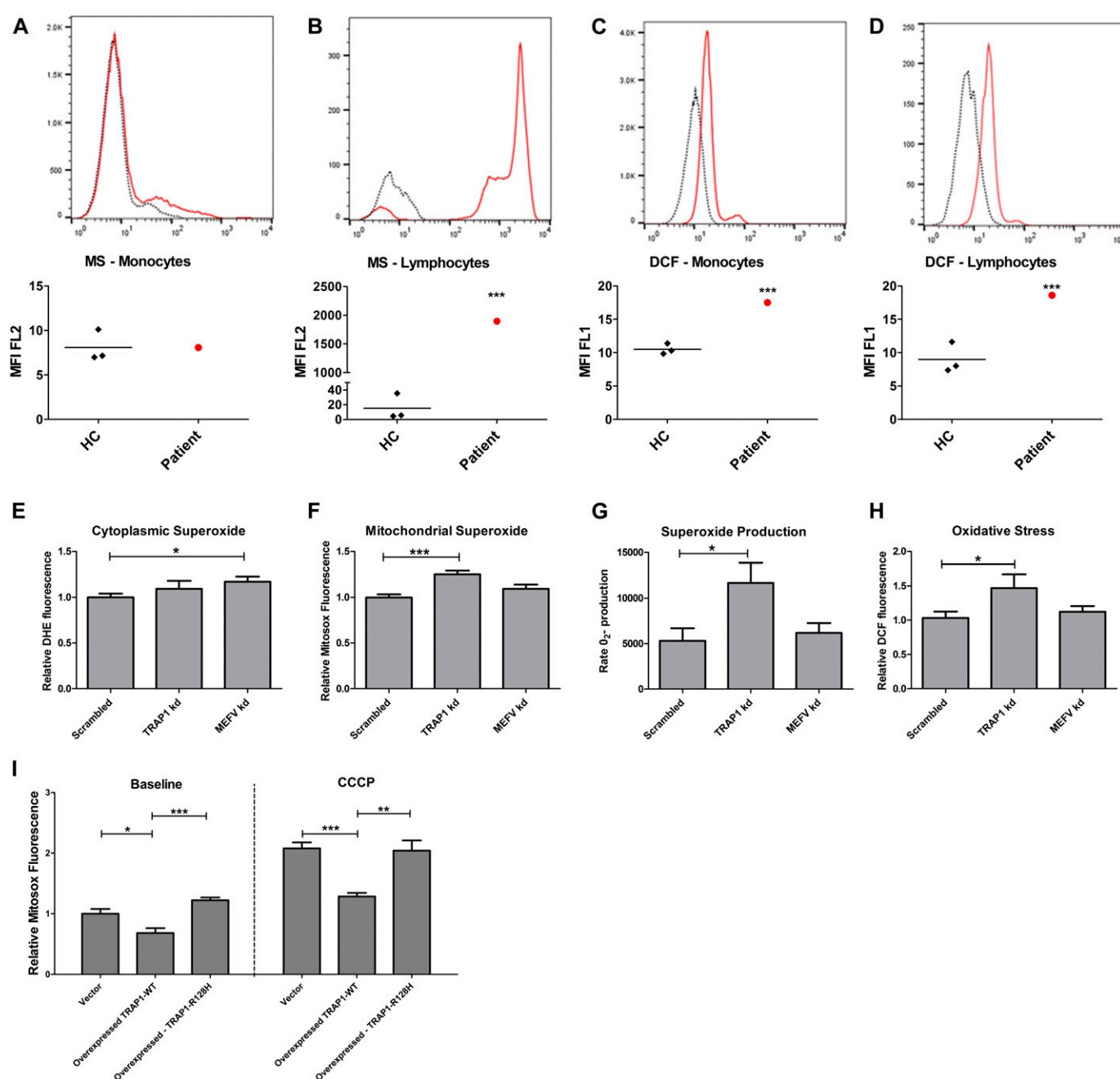

**Figure 2. Flow cytometric and electron spin resonance detection of reactive oxygen species.**
PBMCs isolated from patient 4 (see main text) and 3 healthy controls (HC) were stained for 15 min with 5 $\mu$M MitoSox or 10 $\mu$M DHE and analysed immediately by flow cytometry. Dotted lines in (A, B, C, D) denote HC (one representative experiment); red lines denote patient 4. **(A, B)** Mitochondrial oxide in monocytes (A) showed no significant difference, but (B) was increased in lymphocytes from patient 4 (***$P < 0.001$, two-tailed Z-test). **(C, D)** Oxidative stress detected by H2DCFDA was elevated in both patient monocytes (C) and lymphocytes (D). Fig 2E–H: 500,000 THP1 cells were seeded into a 24-well plate with 100 nM PMA. After 24 h, the medium was replaced and lipofectamine and siRNA complexes were added. After a further 24 h, the medium was replaced with RPMI with 2% FCS for another 24 h. **(E, F, G, H)** Cells were then trypsinized and either stained for 15 min with 5 $\mu$M MitoSox (E), 10 $\mu$M DHE (F), or 10 $\mu$M H2DCFDA (H) and analysed immediately by flow cytometry; or the cells were resuspended in antioxidant buffer, spin probe CMH was then added, and superoxide production was measured 10 times over 10 min via electron spin resonance spectroscopy (G). Cytoplasmic superoxide was significantly increased in MEFV knockdown (kd) cells compared with scrambled control. Mitochondrial superoxide, superoxide production, and oxidative stress levels were significantly increased in the TRAP1 kd cells. **(I)** THP1 cells were transfected with full-length C-terminal DDK-tagged TRAP1 in a pCMV6-Entry vector, with either wild-type TRAP1 sequence or with the same point mutation inducing the p.R128H protein change or the empty vector alone. 500,000 THP1 cells were seeded into a 24-well plate in RPMI with 10% FCS and 100 nM PMA overnight. Relevant wells were treated with 5 $\mu$M CCCP for 6 h. The cells were then stained with 5 $\mu$M MitoSox, trypsinized, and analysed immediately by flow cytometry. At baseline, MitoSox fluorescence was slightly but significantly decreased in only the WT-TRAP1 overexpressing cells compared with the vector control. Upon treatment with CCCP, mitochondrial superoxide increased in vector control and R128H mutant TRAP1, but to a significantly lesser extent in the WT-TRAP1 cells. kd, knockdown; WT, wild-type.

affect superoxide levels, THP1 cells were transfected with full-length C-terminal DDK-tagged *TRAP1* in a pCMV6-Entry vector, with either wild-type *TRAP1* sequence or with the p.R128H point mutation. These cell lines were then differentiated to a macrophage-like phenotype, and mitochondrial superoxide levels were measured by flow cytometry. Carbonyl cyanide m-chlorophenyl hydrazone (CCCP) is an ionophore, which uncouples the electron transport chain in the mitochondria, increasing the rate of production of superoxide. Overexpression of both wild-type and p.R128H mutated *TRAP1* protected the cells from the induction of mROS caused by CCCP, although significantly less-so for the p.R128H mutant protein, indicating that this is a hypomorphic mutation that retains some functionality in this respect (Fig 2I). Overexpression of TRAP1 has been shown to preserve mitochondrial membrane potential in astrocytes (Voloboueva et al, 2007), and it regulates the opening of the mitochondrial permeability transition pore (Hua et al, 2007; Xiang et al, 2010), which might explain this protective effect on membrane potential outside of protein folding.

As TRAP1 may also have a role in protein folding in the ER, we evaluated the level of splicing of X-box protein 1 (XBP1) mRNA, which is increased under conditions of ER stress, in patient 4 compared with three healthy controls. We found that whereas the amount of unspliced *XBP1* (*uXBP1*) was similar, total amount of spliced *XBP1* (*sXBP1*) was significantly elevated in patient 4 ($P < 0.001$), and thus the ratio of *sXBP1*/*uXBP1* was also increased ($P < 0.001$) (Fig 3A). *sXBP1* is then translated to give the functional transcription factor protein. The ER and the mitochondria can form tight associations of their membranes, called mitochondrial associated membranes (MAMs), which allow exchange of signalling events and molecules between them. Upon inflammasome activation, NLRP3 and ASC localize to MAMs, which facilitates detection of ROS production from damaged mitochondria (Zhou et al, 2011; Missiroli et al, 2018). We also performed confocal microscopy looking at the cellular localization of TRAP1 in PBMCs of patient 4 and healthy controls. Interestingly, we found that TRAP1 mainly resides in the ER of healthy PBMCs at rest, whereas in patient 4, the organelles show less definition and a greater degree of association between the mitochondria and the ER and TRAP1 (Fig 3B) potentially indicating the formation of MAMs.

TRAP1 was so named as it was identified in a yeast 2-hybrid screen as binding to tumour necrosis factor receptor 1 (TNFR1); (Song et al, 1995). Given the role of TRAP1 in refolding proteins in both the mitochondria and ER, and that one pathogenic mechanism in the autoinflammatory disease TRAPS may be accumulation of structurally misfolded TNFR1 leading to ER stress, XBP1 splicing, and increased mROS generation (Simon et al, 2010; Bulua et al, 2011; Dickie et al, 2012), we hypothesized that TRAP1 may chaperone the folding of TNFR1 in the ER, resulting in increased TRAP1/TNFR1 co-localization and possible TRAP1 sequestration in patients with TRAPS. To explore this further, we obtained PBMCs from TRAPS patients with a range of mutations in *TNFRSF1A* and determined the cellular localization of TRAP1 and TNFR1 using confocal microscopy (Fig 4A). We found a significantly greater degree of co-localization of these two proteins in patients with structural *TNFR1SFA* mutations affecting cysteine, and in a single patient with a *TNFRSF1A* intron 6 variant (probably also causing major structural protein changes), compared with patients with milder *TNFRSF1A* variants (p.R92Q; p.Y38S; Fig 4B). This suggests a chaperone role for TRAP1 in patients

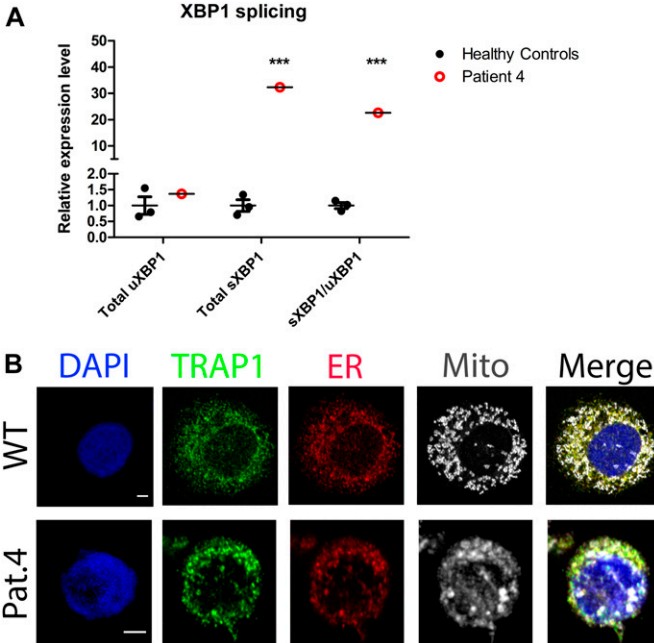

**Figure 3. *XBP1* splicing and TRAP1 localization.**
**(A)** *XBP1* mRNA is spliced during the unfolded protein response under ER stress. *XBP1* splicing was evaluated in patient 4 with qPCR and normalised to the average of three HCs (wild-type WT). There was no significant difference in the level of unspliced *XBP1 (uXBP1)*, but the amount of spliced *XBP1 (sXBP1)* was significantly increased ($P < 0.001$, two-tailed Z-test) and thus the ratio of *sXBP1*/ *uXBP1* was also increased ($P < 0.001$, two-tailed Z-test). **(B)** PBMCs from three HCs and patient 4 were seeded onto polylysine coverslips and incubated with mitotracker deep red to label the mitochondria (grey). These were fixed with methanol and labelled with DAPI (blue), anti-TRAP1 antibodies (green), and anti-calnexin antibodies (red) to stain the ER. Scale bar = 2 $\mu m$. TRAP1 showed localization to the ER in WT cells. In patient 4, the mitochondria showed increased association with the ER and TRAP1 and less structural definition.

with TRAPS, potentially regulating mROS production in response to TNFR1 misfolding. In that context, TNFR1 was measured (by ELISA in serum as part of routine clinical care) on seven discrete occasions for IV-1 (and was elevated in 2/7 samples), and on five discrete occasions in IV-2 (elevated in 3/5 samples). There was no obvious relationship with other inflammatory markers (Table S3). Regarding the clinical deterioration in patient IV-1 after a single dose of infliximab, we noted some similarity with the TRAPS. Infliximab may provoke increased pro-inflammatory cytokine production in some patients with TRAPS (Nedjai et al, 2009); the mechanism of clinical deterioration in IV-1 is, however, unknown.

In addition to the aforementioned serum cytokine studies performed in IV-1 and IV-2 as part of routine care, we investigated the levels of a panel of cytokines and chemokines (IL-1, IL-6, IL-8, IL-10, IL-18, IL-18BP, TNF, INFγ, IP-10, and MCP-1) in the serum of three of the patients: IV-1, IV-2, and patient 4. We found elevated levels of IL-18 in all three patients, which normalised in IV-1 and IV-2 to healthy control levels post-HSCT (Fig 5A). IL-18 is an alarmin, released upon tissue damage to activate the immune system (Blom & Poulsen, 2012; Chan et al, 2012; Soares et al, 2017). IL-18 binding protein (IL-18-BP), which is an antagonist of IL-18, was not elevated (Fig 5B). The levels of the other cytokines tested were within the normal range (data not shown). IL-1, like IL-18, is produced upon inflammasome

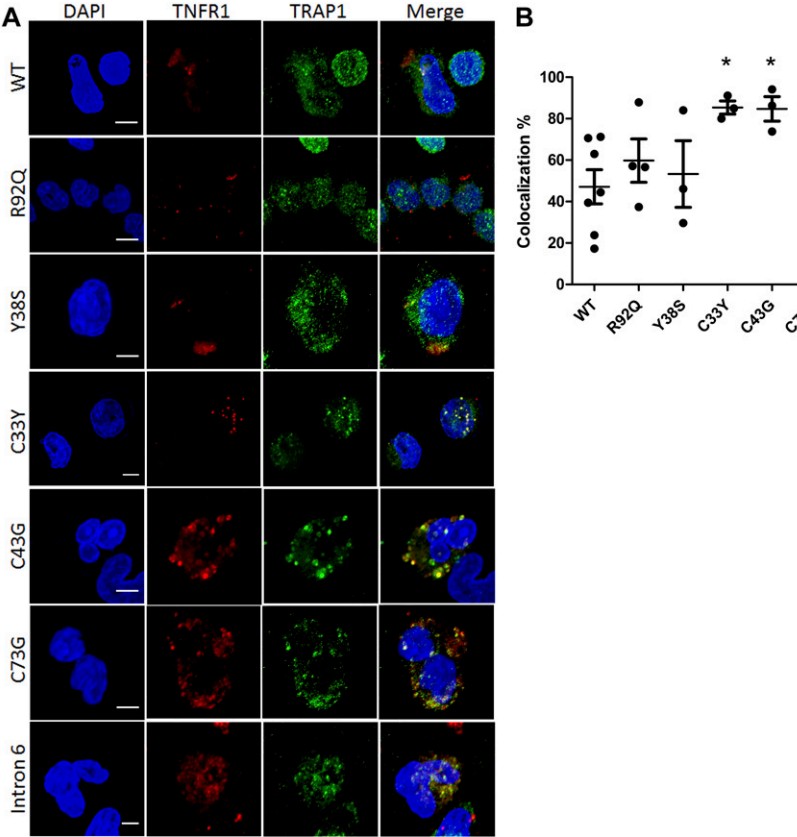

**Figure 4. Confocal microscopy: co-localization of TRAP1 and TNFR1 in PBMCs from patients with TNF receptor associated periodic syndrome (TRAPS). (A)** PBMCs were isolated from TRAPS patients with different mutations and from three HCs (wild-type, WT), fixed, and labelled with the nuclear stain DAPI (blue), anti-TNFR1 antibodies (red), and anti-TRAP1 antibodies (green). Scale bar = 5 $\mu$m. **(B)** The degree of red-green co-localization was quantified in at least three images for each patient and from three HCs (WT). *$P < 0.05$ **$P < 0.01$ (unpaired two-tailed $t$ tests).

and caspase-1 activation, but the former is often difficult to detect in the serum as it acts locally and breaks down quickly. We found that cultured PBMCs from patient 4 after stimulation produced more IL-18 and IL-1 than healthy control cells (Fig 5C) and showed a greater degree of caspase-1 activation (Fig 5D) required for the secretion of both these cytokines. Unfortunately, these experiments were only performed once because of limited availability of cells.

Anakinra has been shown to partially antagonize IL-18 (Brydges et al, 2013; Vastert et al, 2014; Standing et al, 2016); the second sister in the kindred showed partial response, and thus far, patient 4 has shown complete remission with this treatment, but we cannot specify if this is due to IL-1, IL-18, or (perhaps more likely) antagonism of both these pro-inflammatory cytokines. IL-18 and IL-18-BP levels, preceding HSCT, in IV-1 and IV-2 were similar to those we previously reported in patients with the severe autoinflammatory disease periodic fever immunodeficiency and thrombocytopaenia (PFIT), caused by recessive mutation in *WDR1* (Fig 5A and B) (Standing et al, 2016). PFIT is also characterised by periodic fevers, severe oromucosal ulceration, and only partial response to anakinra, ultimately requiring allogeneic HSCT for disease control. We also previously reported abnormal WDR1 intracellular protein aggregates in PFIT (Standing et al, 2016), which could represent misfolded protein with propensity to activate IL-18, thus driving autoinflammation by a similar common pathway as mutated TRAP1.

In summary, we have described a severe autoinflammatory disease in three siblings associated with homozygous *MEFV* p.S208C mutations, and homozygous *TRAP1* p.R128H variants characterised

by recurrent fevers, systemic inflammation, oronasal mucosal inflammation, transient skin rashes, and noninfectious osteitis with fatal outcome in one sibling. Allogeneic HSCT was curative in the two surviving siblings. We suggest that this digenic disease model might have accounted for the particularly severe phenotype in this kindred. Discovery of another unrelated patient with autoinflammation, with the p.R128H *TRAP1* variant in compound heterozygous state with two other rare, predicted damaging *TRAP1* variants, and our finding that the p.R128H *TRAP1* mutation was an important driver of mROS production supports the suggestion that bi-allelic hypomorphic *TRAP1* mutations might contribute to autoinflammation. Our findings could, therefore, suggest a role for TRAP1 as an important protein chaperone, with loss of function leading to cellular stress and an autoinflammatory phenotype, driven by IL-18.

# Materials and Methods

### Ethics

This study was approved by the Bloomsbury ethics committee (ethics number 08H071382); we obtained written informed consent from all the family members, patients, and healthy adult controls who participated. Anonymised paediatric age-matched control sera, surplus to routine clinical requirements, were obtained from the Immunology Laboratory at Great Ormond Street Hospital NHS

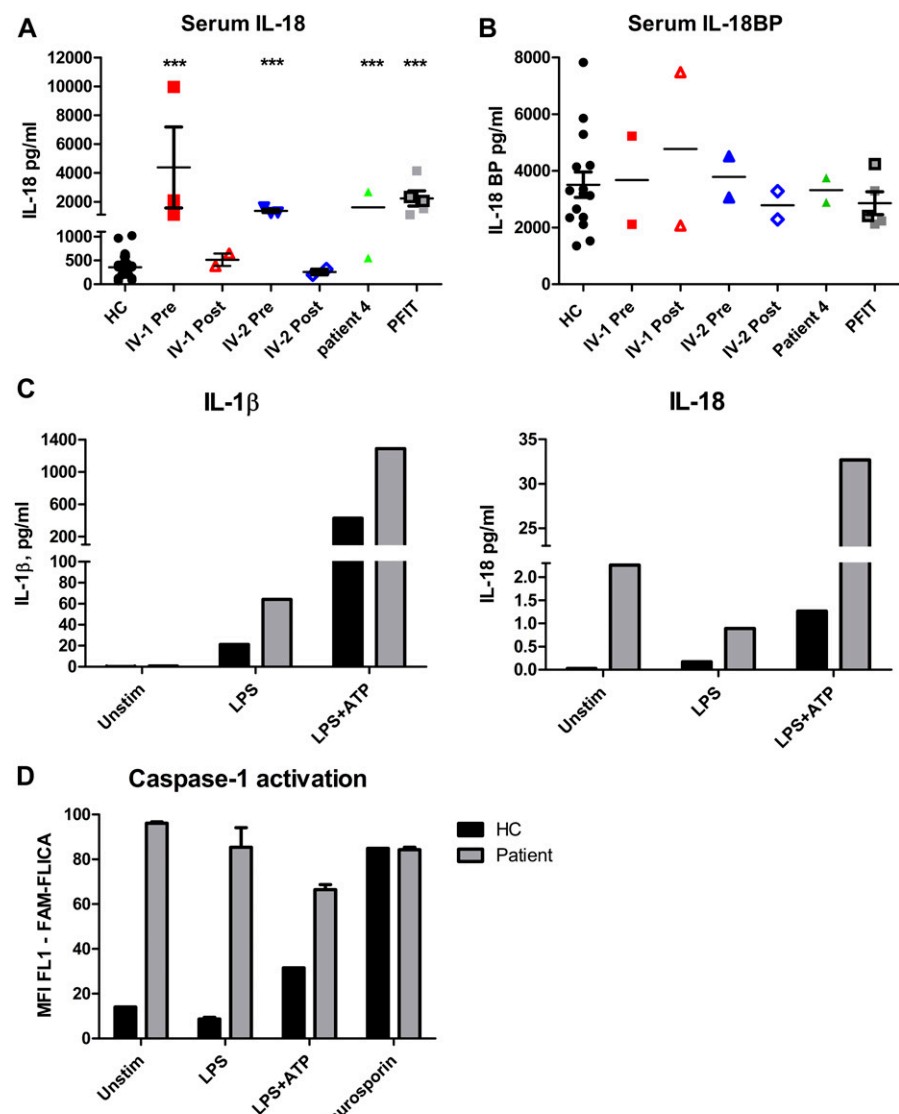

**Figure 5. Serum IL-18 measurements.**
**(A)** Serum IL-18 levels in HCs, patient IV-1, and IV-2 before and after haematopoietic stem cell transplantation and from two patients with the autoinflammatory disease periodic fever immunodeficiency and thrombocytopenia (PFIT; see main text); outlined and grey squares represent repeated measures from two patients at different times. Pre-haematopoietic stem cell transplantation IL-18 levels in IV-1 and IV-2 and patient 4 were significantly higher than those in HCs and similar to PFIT patients; ***$P < 0.001$ (two-tailed unpaired $t$ tests). **(B)** Corresponding IL-18 binding protein (IL-18BP) levels in IV-1, IV-2, and patient 4 which showed no significant difference to levels in HCs or patients with PFIT. **(C)** IL-18 and IL-1 secretion in PBMCs cultured from patient 4 versus an HC, at baseline or with stimulation with 100 ng/ml LPS for 4 h, or LPS for 4 h followed by 5 mM ATP for 15 min. **(D)** Caspase-1 activation measured as the median fluorescence intensity of cells stained with FLICA detected by flow cytometry in patient 4 and an HC. **(C, D)** Patient shown in grey and HC in black. Experiment was only performed once because of limited availability of cells, and thus, statistical analysis was not possible.

Foundation Trust and were used in some experiments without written consent, in accordance with ethical approval for the use of anonymised laboratory samples surplus to routine requirements. All study subjects consented to publication of this report, including clinical images.

## Genetic mapping and sequencing

200 ng genomic DNA from patients, siblings, and the parents was isothermally amplified and enzymatically fragmented before hybridization to Illumina Human610-Quad arrays overnight. These were imaged using the Illumina iScan. The regions of homozygosity were identified using Illumina's Beadstudio with the loss of heterozygosity detector plug-in (version 1.0.3); the minimum number of contiguous homozygous single nucleotide polymorphisms was set to 100.

Targeted capture for the whole 5-Mb region (chr16:1079582-5901163, hg19) was completed using a NimbleGen custom 385k capture array

and sequenced on the Genome Analyzer II (Illumina). The raw sequence data were aligned to the human reference genome using the Burrows-Wheeler Aligner alignment algorithm. Variant calling was with the Genome Analysis ToolKit. Variant annotation was with ANNOVAR and SNPEFF.

Genetic variants were confirmed and familial segregation ascertained by PCR and Sanger sequencing using the following primers to amplify and sequence *TRAP1* exon 4 forward: 5′-TGGGACCCGAGA-CATCAC-3′ and reverse: 5′-TGCAGCTGACCAGATAGATCC-3′. MEFV exon 2 5′-TGTAAAACGACGGCCATAAACGTGGGACAGCTTCATC-3′. 5′-AGGAAA-CAGCTATGACCACGTGCCGGCCAGCCATTCTTTCTC-3′. Sequencing was completed with the AB3730 using the BigDye v3.1 kit (Applied Biosystems).

RNA was extracted from control and patient 4 using TRIzol (Invitrogen) as per the manufacturer's instructions and reverse transcribed with the high-capacity RNA to cDNA kit (Applied Biosystems). TRAP1 transcript was PCR-amplified, using FastStart Taq Polymerase (Roche), and the following primers: forward 5′-CCAGGCCGAGACAAAGAAG-3′;

and reverse 5′-GGCCTCATAGTAGGGTGAGTG-3′. 4 $\mu$l of the resultant PCR product was annealed with 1 $\mu$l pCR2.1-TOPO (Invitrogen). This was transformed into TOP10 chemically competent *Escherichia coli* (Invitrogen) as per the manufacturer's instructions, grown for 1 h in Super Optimal broth with Catabolite repression (SOC) broth and plated onto S-Gal/Luria Broth Agar Blend plates containing 100 $\mu$g/ml ampicillin (Sigma-Aldrich) and grown at 37°C overnight. White colonies were picked, PCR-amplified with the same primers, and sequenced on the AB3730 using the BigDye v3.1 kit (Applied Biosystems).

### PBMC separation

Blood was collected into Falcon tubes (Thermo Fisher Scientific) containing 35 U preservative-free heparin (CP Pharmaceuticals) per 50 ml. This was then mixed with equal volumes of Roswell Park Memorial Institute media (RPMI) and layered onto Lymphoprep (Axis-Shield). This was centrifuged at 800$g$ for 30 min with the brake off. The PBMC layer was transferred to a fresh tube and resuspended in RPMI and spun at 500$g$ for 10 min.

### Confocal microscopy of PBMC

To assess cellular localization of TRAP1 100,000 PBMCs from patient 4, or healthy controls were seeded onto coverslips coated in poly L-lysine and incubated with 500 nM mitotracker deep red 30 min at 37°C. These were fixed and permeabilized with methanol at −20°C for 20 min. Subsequently, nonspecific binding was blocked with donkey serum for 1 h. Coverslips were then incubated with mouse anti-TRAP1 (BD biosciences) and goat anti-calnexin (Santa-Cruz) at 4°C overnight, followed by Alexa Fluor-488 anti-mouse and Alexa Fluor-568 anti-goat secondary antibodies for 1 h. Specificity of TRAP1 antibody staining was confirmed using Western blotting of wild-type (WT) or TRAP1 kd THP1 cell lysates (data not shown).

To assess TRAP1/TNFR1 co-localization, 100,000 PBMCs from TRAPS patients or three healthy controls were seeded onto coverslips coated in poly L-lysine and allowed to settle for 45 min. These were fixed with 4% PFA for 20 min and then permeabilized with 0.5% triton X for 5 min. After this, nonspecific binding was blocked with donkey serum for 1 h. Coverslips were then incubated with mouse anti-TRAP1 (BD biosciences) and rabbit anti-TNFR1 (Santa-Cruz) for 45 min, and subsequently, Alexa Fluor-488 anti-mouse and Alexa Fluor-568 antirabbit secondary antibodies.

These were mounted with Vectashield mounting medium with DAPI and imaged using the LSM 710 confocal microscope (ZEISS) with 63× objective (Plan-Apochromat; numerical aperture, 1.4; working distance, 190 $\mu$m; imaging medium, oil) with Zen2009 software (ZEISS). Deconvolution was completed using Huygens (Scientific Volume Imaging B.V.) and co-localization analysis was completed using Imaris software (Bitplane), and as a percentage of the red (TNFR1) channel material which co-localized with green (TRAP1). This was compared with wild-type using the $t$ test.

### siRNA transfection

500,000 THP1 cells with 100 nM PMA were seeded per well in a 24-well plate with 0.5 ml RPMI with 10% FCS without antibiotics. The following day, the media was changed on the THP1 cells to remove the PMA. Silencer select predesigned siRNA *TRAP1* (s177) and *MEFV* (s5660), and scrambled control was obtained from Life Technologies and resuspended to 50 $\mu$M in nuclease-free water. 1 $\mu$l of this stock was diluted in 49 $\mu$l of optiMEM (Invitrogen) for each well. 1 $\mu$l of Lipofectamine 2000 (Invitrogen) was diluted in 49 $\mu$l of optiMEM per well. The working solutions of Lipofectamine and RNA were mixed in equal volumes and incubated for 20 min to permit the formation of transfection complexes. 100 $\mu$l was then added to each well containing the cells. The following day, the media was changed to RPMI with 2% FCS. After a further 24 h, the cells were ready for further experiments.

### Overexpression of *TRAP1*

1 million THP1 cells were transfected with 0.5 $\mu$g of pCMV6-AC-DKK vector (OriGene) containing wild-type or R128H TRAP1, or empty vector using the Amaxa nucleofector following the manufacturer's protocol for kit V. Mutated TRAP1 was produced using Q5-site directed mutagenesis kit (NEB) with the following primers, following the manufacturer's instructions. 5′-GAAAAACTGCATCACAAACTGGTG-3′ and 5′-CAAGGCATCGCTGGCATT-3′.

### Flow cytometry for ROS detection

Cells were incubated in 10 $\mu$M dichlorodihydrofluorescein diacetate (H2DCFDA), 10 $\mu$M dihydrothidium (DHE), or 5 $\mu$M MitoSOX (mitochondrially targeted DHE) in phenol-free RPMI at 37°C for 30 min. They were washed and placed on ice and before immediate analysis by flow cytometry.

### Electron spin resonance

Krebs-Hepes buffer was prepared using ultrapure chemicals from Sigma-Aldrich with iron and copper concentrations below five parts per million. The Krebs-Hepes buffer was filtered through a 0.22-$\mu$M filter and the pH adjusted to 7.4. Antioxidant buffer was prepared by adding 25 $\mu$M of deferoxamine (DFO) and 5 $\mu$M diethyldithiocarbamate (DETC). Spin probe solution was made by adding 10 mM of 1-hydroxy-3-methoxycarbonyl-2,2,5,5-tetramethylpyrrolidine (CMH) to antioxidant buffer. Cells to be analysed were trypsinized and resuspended in 45 $\mu$l of antioxidant buffer on ice. Immediately before analysis, 2.5 $\mu$l of spin probe solution was added, the cells were drawn into the capillary tube, and placed in the E-SCAN holder in the temperature controller at 37°C. The samples were analysed using the E-SCAN using the settings in Table 1.

### XBP1 splicing qPCR

For XBP1 splicing analysis, PBMC cDNA from patient 4 and three healthy controls was measured using quantitative PCR with the standard curve method. qPCR was performed with the RotorGene 6000 machine (Corbett) and SYBR green JumpStart Taq ReadyMix (Sigma-Aldrich) following the manufacturers recommendations and using the following primers. Spliced XBP1 forward: 5′-CTGAGTCCG-CAGCAGGTG-3′, unspliced XBP1 forward: 5′-TCCGCAGCACTCAGACTACG-3′ and XBP1 reverse: 5′-AGTTGTCCAGAATGCCCAACA-3′. HPRT housekeeping control forward: 5′-GGAAAGAATGTCTTGATTGTGGAAG-3′ and reverse: 5′-GGATTATACTGCCTGACCAAGGAA-3′.

**Table 1.** Settings for the E-SCAN electron spin resonance spectrometer.

| | |
|---|---|
| Centre field | 1.99$g$ |
| Microwave power | 20 mW |
| Modulation amplitude | 2 G |
| Sweep time | 10 s |
| Number of scans | 10 |
| Field sweep | 60 G |

### Measurement of cytokines

IL-18 was measured using a Meso Scale Discovery multiplex kit (Meso Scale Diagnostics), as per the manufacturer's instructions in patients and control sera. Control sera was obtained from healthy paediatric controls obtained for the development of diagnostic tests for primary immunodeficiency (Research Ethics Committee reference 06/Q0508/16; n = 40; 18 males; median age, 5 yr; range, 0.3–17 yr).

### Statistical analyses

Unpaired $t$ tests and graphs were produced using Prism version 4 (GraphPad). Z-tests were completed in excel. $P < 0.05$ was considered significant.

## Supplementary Information

## Acknowledgements

We would like to thank Frank O'Neill and Eve McLoughlin for help with the electron spin resonance assay, and Stefano Masi for help with protocols for flow cytometric detection of reactive oxygen species. All research at Great Ormond Street Hospital NHS Foundation Trust and UCL Great Ormond Street Institute of Child Health is made possible by the NIHR Great Ormond Street Hospital Biomedical Research Centre. The views expressed are those of the author(s) and not necessarily those of the NHS, the NIHR or the Department of Health.

### Author Contributions

ASI Standing: conceptualization, data curation, formal analysis, funding acquisition, investigation, methodology, project administration, and writing—original draft, review, and editing.
Y Hong: conceptualization, supervision, investigation, methodology, and writing—review and editing.
C Paisan-Ruiz: conceptualization, data curation, formal analysis, supervision, investigation, methodology, and writing—review and editing.
E Omoyinmi: data curation, investigation, and writing—review and editing.
A Medlar: formal analysis.
H Stanescu: formal analysis.
R Kleta: formal analysis and writing—review and editing.
D Rowczenio: formal analysis, investigation, and writing—review and editing.
P Hawkins: investigation and writing—review and editing.
H Lachmann: investigation and writing—review and editing.
MF McDermott: investigation and writing—review and editing.
D Eleftheriou: conceptualization, data curation, formal analysis, supervision, investigation, methodology, and writing—review and editing.
N Klein: conceptualization and writing—review and editing.
PA Brogan: conceptualization, data curation, supervision, funding acquisition, investigation, methodology, and writing—review and editing.

### Conflict of Interest Statement

The authors declare that the research was conducted in the absence of any commercial or financial relationships that could be construed as a potential conflict of interest.

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
