## [Reviewer comments · Life Science Alliance]

Life Science Alliance

TRAP1 chaperone protein mutations and autoinflammation

Ariane Standing, Ying Hong, Coro Paisan-Ruiz, Ebum Omoyinmi, Alan Medlar, Horia Stanescu, Robert Kleta, Dorota Rowczenio, Philip Hawkins, Helen Lachmann, Nigel Klein, Michael McDermott, Despina Eleftheriou, and Paul Brogan

DOI: <https://doi.org/10.26508/lsa.201900376>

Corresponding author(s): Ariane Standing, UCL and Great Ormond Street Institute of Child Health

Review Timeline:

Submission Date:	2019-03-12
Editorial Decision:	2019-04-08
Revision Received:	2019-08-14
Editorial Decision:	2019-09-20
Revision Received:	2019-12-19
Accepted:	2019-12-20

Scientific Editor: Andrea Leibfried

Transaction Report:

April 8, 2019

Re: Life Science Alliance manuscript #LSA-2019-00376-T

Dr. Ariane Shamilla Isaura Standing
UCL Institute of Child Health
III
30 Guildford Street
London WC1N 1EH
United Kingdom

Dear Dr. Standing,

Thank you for submitting your manuscript entitled "Hypomorphic TRAP1 mutations enhance autoinflammatory phenotypes, identifying a key protein chaperone in inflammation" to Life Science Alliance. The manuscript was assessed by expert reviewers, whose comments are appended to this letter.

As you will see, both reviewers appreciate your findings but think that your conclusions are not sufficiently supported by the data provided. They provide constructive input on how to lend more support to your conclusions by including a defined set of additional controls and experiments. These seem all straightforward to perform and we would therefore like to invite you to submit such a revised manuscript to us. The requested clarifications / text changes should get addressed, too.

Thank you for this interesting contribution to Life Science Alliance. We are looking forward to receiving your revised manuscript.

Sincerely,

B. MANUSCRIPT ORGANIZATION AND FORMATTING:

Reviewer #1 (Comments to the Authors (Required)):

Review 26th April 2017

- You should be trying to help the work get published not necessarily in this journal but ultimately.
- Don't criticize an experiment unless you can tell the authors how they could do it better. "If you just want to throw darts," he would say, "go to the pub."
- Keep in mind that no one ever built a statue to a critic.
- Try to act as a peer in the process of peer review.

Science Signaling 2009 Michael Yaffe

Title: Hypomorphic TRAP1 mutations enhance autoinflammatory phenotypes, identifying a key protein chaperone in inflammation

Manuscript # LSA-2019-00376-T

General Remarks

This study describes patients with both hypomorphic PYRIN mutations and a TRAP1 mutation that results in a severe inflammatory disease, much more severe than is typically seen in the S208 mutant alone. Fig. 1 describes the clinical phenotype and response to treatment, Fig. 2 looks at mitochondrial function in cells from these TRAP1 mutant patients and also in cell lines with TRAP1 knock-down. Fig. 3 looks at cells from TNFR1 mutant TRAPS patients and shows an increased co-localisation with the more severe TNFR1 mutants. This is quite a logical leap. As far as I understand the authors wish to show that their R128H mutant is a hypomorph, therefore why do they not look at TNFR1 levels in these patient samples? Another point that is concerning to me is that the R128H variant is present at 0.3% in the "general" population (and 3 homozygotes) according to ExAC, furthermore there are R128C and R128L mutants also present. Probably because of the small numbers there is no statistical analysis of whether therefore this variant could occur by chance. It would be interesting to see the frequency of this variant in other unconnected inflammatory diseases.

In addition to these issues I think that for this manuscript to have some scientific rigour the authors should show that their TRAP1 antibody staining in Fig. 3 is specific.

Finally the statement that "Infliximab binds to the extracellular component of transmembrane TNFR1 and causes a series of outside-in signalling effects, culminating in increased intracellular ROS and apoptosis, (Horiuchi et al., 2010)" is absolutely wrong, as in, 1. this review did not say this and 2. neither does it happen. Infliximab binds TNF not TNFR1. Therefore the hypothesis put forward in the following sentence is undoubtedly incorrect. And incidentally the evidence for reverse TNF signalling is tenuous.

Reviewer #2 (Comments to the Authors (Required)):

Comments on 'Hypomorphic TRAP1 mutations enhance autoinflammatory phenotypes, identifying a key protein chaperone in inflammation.'

Standing et al. describe in this manuscript the autoinflammatory phenotype of a kindred with a known pathogenic mutation in MEFV (p.S208C). Given the unexpected severe phenotype, the authors suspect an oligogenic cause underlying the phenotype. Further genetic research reveals the presence of a second homozygous mutation present in TRAP1. Given its role as a chaperone of TNFR1 and the known autoinflammatory condition caused by germline mutations in TNFR1 (TRAPS), the authors hypothesize that this mutation could underlie the more severe phenotype.

To test this hypothesis, the authors rely on the identification of an unrelated patient suffering from autoinflammation with no known variants in genes causing autoinflammatory syndromes but with a compound heterozygous mutation in TRAP1. The main argument for this strategy relies on the fact that Gall patients from the original family have been successfully transplanted, fully ablating cardinal symptoms of autoinflammation.

In a first series of experiments, the authors document increased (total and mitochondrial) ROS productions in primary lymphocytes, monocytes, the monocyte derived cell-line THP-1 in two different assays.

In a second series of experiments, the authors test the hypothesis that defective TNFR1 folding in the ER results in UPR, which might contribute to the development of TRAPS. The colocalization of TRAP-1 and TNFR1 is tested in primary PBMCs from patients carrying known mutations in TNFR1. These experiments reveal that some variants of TNFR1 are indeed associated with increased colocalization.

Finally, by measuring cytokines in the serum of the patients carrying the TRAP1 variants, the authors note increased IL-18 concentrations and speculate that protein aggregates due to the TRAP1 variant could result in IL-18 activation.

General comment:

Standing et al. propose a working model in which TRAP1 variants result in IL-18 activation by increased ROS production due to protein aggregates in the ER. This augments autoinflammation in one kindred or induce autoinflammation in its own right in a fourth patient.

The manuscript is clear and well written. The experimental data seem well controlled aside of some minor remarks. One major comment is that the data is mainly showing associations rather than formal prove. In its current state, the manuscript lacks some key experiments to substantiate the proposed model.

Particular questions that should be addressed:

1. Does the observed TRAP1 variant result in protein aggregates in the ER (of TNFR1) in a similar fashion as the described TNFR1 variants in this study or the WDR1 aggregates in PFIT?
2. Does the observed TRAP1 variant result in UPR activation?
3. Can the phenotype in primary PBMCs be restored? Does overexpression of the WT TRAP1 (ideally with MT TRAP1 as a control) in the PBMCs of the unrelated patient normalizes ROS production and/or reduce protein aggregates and/or reduce IL-18 production?

Specific comments:

1. Page 1: '...and characterises a new autoinflammatory disease caused by bi-allelic mutations in TRAP1.' At present, the manuscript does not contain sufficient evidence to establish the causal relation between TRAP1 and the autoinflammatory syndrome.

This sentence should be amended.

2. Page 3: leads to increased ROS generation, accumulation of peroxyated proteins, and autoinflammation associated with elevated levels of serum IL-18.

a. To my knowledge peroxylation of proteins has not been shown.

b. It has not formerly been proven that elevated levels of IL-18 are the result of the TRAP1 variant
This sentence should be amended.

3. Page 3: '...routinely available genetic screening for the common causes of autoinflammatory disease revealed wild type for MVK, TNFRSF1A, NLRP3, and MEFV (exons 2 and 10).' This sentence seems to be inconsistent with page5 in which a mutation in MEFV exon 2 is described.
Please amend.

4. Page 5: Please also describe the allele frequency of the TRAP1 variant. Analysis of the Gnomad

database reveals the increased prevalence of the particular variant in South Asia (0,01) compared to other regions and the identification of 6 homozygote carriers suggesting the presence of incomplete penetrance.

5. Page 6: It is unclear why dermal fibroblasts from the original kindred were not considered to measure mitochondrial and total ROS production.

6. Page 6: The authors measure increased mtROS production in lymphocytes. At present it is not clear whether this represents comparable lymphocyte populations. Given the large body of evidence that T cell populations undergo fundamental metabolic rewiring upon activation, this piece of evidence showing altered mtROS production in lymphocytes could also be interpreted as a consequence (rather than a cause) of the autoinflammatory syndrome. Activated lymphocyte population (e.g. CD45RO+ cells) should be compared or this alternative possibility should at least be discussed.

7. Page 7: 'This rate of O₂⁻ production was sufficient to induce significant cellular oxidative stress, with an increase in peroxyated proteins, detected as an increase in DCF fluorescence following TRAP1 kd, but not MEFV kd (Fig. 2H).' Peroxyated protein content has not formally quantified, please amend.

8. Page 8: It is an intriguing finding that overexpression of TRAP1, mainly described to be involved in protein folding, can reduced the superoxide formation upon uncoupling the electron transport chain. What is the proposed working mechanism of this observation?

9. Page 8: '...that one pathogenic mechanism in the autoinflammatory disease TRAPS may be accumulation of structurally misfolded TNFR1 leading to ER stress and increased mROS generation'. Showing any indication of UPR activation would be of interest. Also the role of Mitochondria-Associated Membranes (MAMS) could be discussed.

10. Page 8: How do the authors pair a sole increase of IL-18 in the unrelated patient with the complete response to anakinra as described on page 6.

11. Page 9: 'Our findings therefore define a role for TRAP1 as an important protein chaperone regulating mitochondrial and ER protein misfolding, with their loss of function leading to cellular stress and an autoinflammatory phenotype, driven by IL-18.' This sentence should be amended: protein misfolding has not formally been proven.

12. Page 10: Fig 1H. Is there any indication of haploinsufficiency?

13. Page 13: Fig 2E-H. The figures do not match with the legend. Please amend.

14. Page 13: kd should be written in full at the end of legend (similar to other legends).

15. Page 14: It is unclear from the legend or the material and methods how colocalization was quantified and differences were tested statistically. Furthermore, given that only 3 images were used, showing the independent values rather than error bars is indicated.

16. Page 15: PFIT should be written in full at the end of legend (similar to other legends).

17. Page 16: Primer sequences should be added.

Reviewer #1 (Comments to the Authors (Required)):

Title: Hypomorphic TRAP1 mutations enhance autoinflammatory phenotypes, identifying a key protein chaperone in inflammation

Manuscript # LSA-2019-00376-T

General Remarks

This study describes patients with both hypomorphic PYRIN mutations and a TRAP1 mutation that results in a severe inflammatory disease, much more severe than is typically seen in the S208 mutant alone. Fig. 1 describes the clinical phenotype and response to treatment, Fig. 2 looks at mitochondrial function in cells from these TRAP1 mutant patients and also in cell lines with TRAP1 knock-down. Fig. 3 looks at cells from TNFR1 mutant TRAPS patients and shows an increased co-localisation with the more severe TNFR1 mutants. This is quite a logical leap. As far as I understand the authors wish to show that their R128H mutant is a hypomorph, therefore why do they not look at TNFR1 levels in these patient samples?

We did indeed do this (at the time, as part of routine clinical care) and now include the results as a supplementary table, with a brief statement – as can be seen in the table S3, circulating TNFR1 was measured (by ELISA in serum in the routine immunology lab; also other cytokines routinely measured at the time) on 7 occasions for IV-1 (was elevated in 2/7 samples); and 5 occasions in IV-2 (elevated in 3/5 samples). There was no obvious relationship with other inflammatory markers, but nonetheless these data could support the mechanism we propose. No sample was available from IV-4.

Another point that is concerning to me is that the R128H variant is present at 0.3% in the "general" population (and 3 homozygotes) according to ExAC, furthermore there are R128C and R128L mutants also present. Probably because of the small numbers there is no statistical analysis of whether therefore this variant could occur by chance. It would be interesting to see the frequency of this variant in other unconnected inflammatory diseases.

Thanks for this comment. Outside of this study, we have performed next generation sequencing as part of routine care of this gene in 315 patients with suspected autoinflammation; the TRAP1 p.R128H variant was only detected in heterozygous state in one patient (patient 4, now included in this study), emphasising that this variant is rare

even in a cohort of patients with suspected autoinflammatory disease. We have now added a brief statement to this effect in the manuscript.

In addition to these issues I think that for this manuscript to have some scientific rigour the authors should show that their TRAP1 antibody staining in Fig. 3 is specific.

We did indeed confirm the specificity of the TRAP1 antibody with Western blot below Fig 1, using cell lysates of THP1 cells.

Figure 1: Western blot of cell lysates with TRAP1 antibody. Lane 1: Recommended positive control, HeLa cell lysate from BD transduction laboratories, 2-4 and 8, THP1 cell lysates transfected with negative control siRNA. 5-7,9 THP1 cells transfected with TRAP1 siRNA (TRAP1 knockdown).

Finally the statement that "Infliximab binds to the extracellular component of transmembrane TNFR1 and causes a series of outside-in signalling effects, culminating in increased intracellular ROS and apoptosis, (Horiuchi et al., 2010)" is absolutely wrong, as in, 1. this review did not say this and 2. neither does it happen. Infliximab binds TNF not TNFR1. Therefore the hypothesis put forward in the following sentence is undoubtedly incorrect. And incidentally the evidence for reverse TNF signalling is tenuous.

We apologize for our lack of clarity- we in fact should have referred to the binding of infliximab to the complex formed by TNF (free and transmembrane bound) to TNFR1. We have corrected this now. We definitely acknowledge the speculative nature of our statement, and have amended it accordingly.

Reviewer #2 (Comments to the Authors (Required)):

Comments on ' Hypomorphic TRAP1 mutations enhance autoinflammatory phenotypes, identifying a key protein chaperone in inflammation.'

Standing et al. describe in this manuscript the autoinflammatory phenotype of a kindred with a known pathogenic mutation in MEFV (p.S208C). Given the unexpected severe phenotype, the authors suspect an oligogenic cause underlying the phenotype. Further genetic research reveals the presence of a second homozygous mutation present in TRAP1. Given its role as a chaperone of TNFR1 and the known autoinflammatory condition caused by germline mutations in TNFR1 (TRAPS), the authors hypothesize that this mutation could underlie the more severe phenotype.

To test this hypothesis, the authors rely on the identification of an unrelated patient suffering from autoinflammation with no known variants in genes causing autoinflammatory syndromes but with a compound heterozygous mutation in TRAP1. The main argument for this strategy relies on the fact that Gall patients from the original family have been successfully transplanted, fully ablating cardinal symptoms of autoinflammation.

In a first series of experiments, the authors document increased (total and mitochondrial) ROS productions in primary lymphocytes, monocytes, the monocyte derived cell-line THP-1 in two different assays.

In a second series of experiments, the authors test the hypothesis that defective TNFR1 folding in the ER results in UPR, which might contribute to the development of TRAPS. The colocalization of TRAP-1 and TNFR1 is tested in primary PBMCs from patients carrying known mutations in TNFR1. These experiments reveal that some variants of TNFR1 are indeed associated with increased colocalization.

Finally, by measuring cytokines in the serum of the patients carrying the TRAP1 variants, the authors note increased IL-18 concentrations and speculate that protein aggregates due to the TRAP1 variant could result in IL-18 activation.

General comment:

Standing et al. propose a working model in which TRAP1 variants result in IL-18 activation by increased ROS production due to protein aggregates in the ER. This augments autoinflammation in one kindred or induce autoinflammation in its own right in a fourth patient.

The manuscript is clear and well written. The experimental data seem well controlled aside of some minor remarks. One major comment is that the data is mainly showing associations rather than formal prove. In its current state, the manuscript lacks some key experiments to substantiate the proposed model.

Particular questions that should be addressed:

1. Does the observed TRAP1 variant result in protein aggregates in the ER (of TNFR1) in a similar fashion as the described TNFR1 variants in this study or the WDR1 aggregates in PFIT?

Following confocal microscopy of patient 4, and 3 healthy controls, we observe that TRAP1 is resident in the endoplasmic reticulum in the healthy controls. In patient 4 there is an increase in the overlap of ER, TRAP1 and the mitochondria, potentially representing the formation of MAMs as suggested by this reviewer, below. We have included this data as figure 3B. We do not see the same formation of protein aggregates, we think it is possible in TRAPS the protein aggregates sequester TRAP1 from other functions as it attempts to refold misfolded protein clients, whereas in these patients it is the TRAP1 itself which is impaired. It would be interesting to follow this up in future studies.

2. Does the observed TRAP1 variant result in UPR activation?

We measured the degree of X Box Protein 1 splicing via qPCR in patient 4 and 3 healthy controls. Under conditions of ER stress XBP-1 mRNA is spliced, which is transcribed to the active form of this transcription factor. We found that the relative total amount of the spliced XBP-1 mRNA was increased 30 fold, with the relative ratio of spliced to unspliced message increasing 20 fold. The amount of unspliced XBP-1 remained relatively constant in patient 4 compared to healthy controls. We have included this data in a new figure 3A.

3. Can the phenotype in primary PBMCs be restored? Does overexpression of the WT TRAP1 (ideally with MT TRAP1 as a control) in the PBMCs of the unrelated patient normalizes ROS production and/or reduce protein aggregates and/or reduce IL-18 production?

Unfortunately no more cells were available (after doing the key experiments asked for above) to conduct this experiment (this is an adult patient, not at our institution); we also suggest, in any case, that this wouldn't necessarily add much more to our observation of rescue of THP1 kd by TRAP1 overexpression.

Specific comments:

1. Page 1: '...and characterises a new autoinflammatory disease caused by bi-allelic mutations in TRAP1.' At present, the manuscript does not contain sufficient evidence to establish the causal relation between TRAP1 and the autoinflammatory syndrome. This sentence should be amended.

We have amended this sentence, with a much softened conclusion.

2. Page 3: leads to increased ROS generation, accumulation of peroxyated proteins, and autoinflammation associated with elevated levels of serum IL-18.

a. To my knowledge peroxylation of proteins has not been shown. b. It has not formerly been proven that elevated levels of IL-18 are the result of the TRAP1 variant

This sentence should be amended.

We have amended this accordingly.

3. Page 3: '...routinely available genetic screening for the common causes of autoinflammatory disease revealed wild type for MVK, TNFRSF1A, NLRP3, and MEFV (exons 2 and 10).' This sentence seems to be inconsistent with page5 in which a mutation in MEFV exon 2 is described. Please amend.

To clarify, at the time, routine testing of *MEFV* only included exon 10 and part of exon 2 (in those pre NGS days, these areas were considered the mutation hot-spots); *MEFV* p.S208C is (unfortunately) in a part of exon 2 which was not sequenced routinely in our clinical laboratory setting. This mutation was thus only unearthed in the context of this study using NGS and Sanger sequencing of the whole *MEFV* gene. The good news is that one result of our research is that all patients now get full *MEFV* gene sequencing (either by NGS or Sanger). We have amended the manuscript to make this clearer.

4. Page 5: Please also describe the allele frequency of the TRAP1 variant. Analysis of the Gnomad database reveals the increased prevalence of the particular variant in South Asia (0,01) compared to other regions and the identification of 6 homozygote carriers suggesting the presence of incomplete penetrance.

Addressed in comment above to reviewer 1, and manuscript amended accordingly.

5. Page 6: It is unclear why dermal fibroblasts from the original kindred were not considered to measure mitochondrial and total ROS production.

The reason is the affected patients did not provide consent for us to do so, even though our ethical approval permitted. The patients are now discharged to adult care, and well, so it is not appropriate to go back to them with another request, unfortunately.

6. Page 6: The authors measure increased mtROS production in lymphocytes. At present it is not clear whether this represents comparable lymphocyte populations. Given the large body of evidence that T cell populations undergo fundamental metabolic rewiring upon activation, this piece of evidence showing altered mtROS production in lymphocytes could also be interpreted as a consequence (rather than a cause) of the autoinflammatory syndrome. Activated lymphocyte population (e.g. CD45RO+ cells) should be compared or this alternative possibility should at least be discussed.

We thank the reviewer for this comment; we did not look at specific lymphocyte subsets, and have thus added a commentary highlighting this limitation as suggested by the reviewer.

7. Page 7: 'This rate of O₂⁻ production was sufficient to induce significant cellular oxidative stress, with an increase in peroxyated proteins, detected as an increase in DCF fluorescence following TRAP1 kd, but not MEFV kd (Fig. 2H).' Peroxyated protein content has not formally quantified, please amend.

Amended accordingly.

8. Page 8: It is an intriguing finding that overexpression of TRAP1, mainly described to be involved in protein folding, can reduced the superoxide formation upon uncoupling the electron transport chain. What is the proposed working mechanism of this observation?

Overexpression of TRAP1 has been shown to preserve mitochondrial membrane potential in astrocytes (Voloboueva et al., 2007), and also regulates the opening of the mitochondrial permeability transition pore (Hua et al., 2007; Xiang et al., 2010), which may explain this protective effect on membrane potential outside of protein folding. We have added this into the manuscript.

9. Page 8: '...that one pathogenic mechanism in the autoinflammatory disease TRAPS may be accumulation of structurally misfolded TNFR1 leading to ER stress and increased mROS generation'. Showing any indication of UPR activation would be of interest. Also the role of Mitochondria-Associated Membranes (MAMS) could be discussed.

We have demonstrated UPR activation (included as fig. 3A) as discussed above. We have also added in some discussion of MAMs and looked at the ER and mitochondrial localization with confocal microscopy.

10. Page 8: How do the authors pair a sole increase of IL-18 in the unrelated patient with the complete response to anakinra as described on page 6.

We have added a sentence discussing this. In essence, we don't know for sure, but others have speculated that anakinra might also impact on IL18, but it is very unclear if this is a true specific targeted antagonism, or more probably a non-specific anti-inflammatory effect of anakinra.

11. Page 9: 'Our findings therefore define a role for TRAP1 as an important protein chaperone regulating mitochondrial and ER protein misfolding, with their loss of function leading to cellular stress and an autoinflammatory phenotype, driven by IL-18.' This sentence should be amended: protein misfolding has not formally been proven. Amended accordingly.

12. Page 10: Fig 1H. Is there any indication of haploinsufficiency?

The parents and 2 unaffected siblings were formally assessed by PB. Detailed past medical history and exam did not indicate any clinical phenotype. In addition, we measured SAA and CRP in the parents and both unaffected siblings and did not find any indication of subclinical inflammation, suggesting no phenotype associated with haploinsufficiency. We have added this to the text.

13. Page 13: Fig 2E-H. The figures do not match with the legend. Please amend. Corrected.

14. Page 13: kd should be written in full at the end of legend (similar to other legends). Amended.

15. Page 14: It is unclear from the legend or the material and methods how colocalization was quantified and differences were tested statistically. Furthermore, given that only 3 images were used, showing the independent values rather than error bars is indicated.

Details have been added (T test added to legend; added software used for co-localization in methods; changed bar chart to dot plot).

16. Page 15: PFIT should be written in full at the end of legend (similar to other legends). This has been amended.

17. Page 16: Primer sequences should be added. We have now included this.

September 20, 2019

Re: Life Science Alliance manuscript #LSA-2019-00376-TR

Dr. Ariane Shamilla Isaura Standing
UCL and Great Ormond Street Institute of Child Health
III
30 Guildford Street
London WC1N 1EH
United Kingdom

Dear Dr. Standing,

Thank you for submitting your manuscript entitled "TRAP1 chaperone protein mutations enhance autoinflammatory phenotypes" to Life Science Alliance. The manuscript was assessed by the original reviewers again, whose comments are appended to this letter.

As you will see, the reviewers appreciate the revision performed but think that a few remaining concerns need addressing. We would thus like to invite you to submit a final version of your manuscript, addressing the points still raised by the reviewers. Additionally, please also:

- indicate the corresponding author in the manuscript file
- add a COI statement
- provide information on the patients' consent for publication of pictures
- There is a callout for Fig 1J in the manuscript text but no according panel in Fig1 - please fix
- add scale bars to Fig3B and 4A

Thank you for this interesting contribution to Life Science Alliance. We are looking forward to receiving your revised manuscript.

Sincerely,

B. MANUSCRIPT ORGANIZATION AND FORMATTING:

Reviewer #1 (Comments to the Authors (Required)):

Review 19th September 2019

- You should be trying to help the work get published not necessarily in this journal but ultimately.
 - Don't criticize an experiment unless you can tell the authors how they could do it better. "If you just want to throw darts," he would say, "go to the pub."
 - Keep in mind that no one ever built a statue to a critic.
 - Try to act as a peer in the process of peer review.
- Science Signaling 2009 Michael Yaffe

Title: TRAP1 chaperone protein mutations enhance autoinflammatory phenotypes

Manuscript # LSA-2019-00376-TR

General Remarks

"we in fact should have referred to the binding of infliximab to the complex formed by TNF (free and transmembrane bound) to TNFR1." This statement is also wrong. Infliximab binds to free TNF, it prevents TNF binding to the receptor, that's how it works, see DOI 10.1074/jbc.M112.433961. It does not bind to TNFR1 in any shape or form. I understand that the reference cited, Nedjai et al, hypothesised this happened but it does not.

The manuscript still has a yawning logical disconnect. Sure TNFR1 mutations cause TRAPs accumulation, but isn't the point that the TRAPs mutation affects TNFR1 levels? Why is this not tested? To me increased levels of TNFR1 in the serum of patients suggest that there is NOT a problem with TNFR1 production and presentation to the membrane, however, at best, this is a surrogate test. The direct test is to simply look at TNFR1 expression on the membrane using flow cytometry or IHC, as for example in Figure 4. If necessary the TRAPs mutants could be over expressed although patient 4 samples would be best.

The Western blot looks encouraging however to control the specificity of the antibody in an immunofluorescence experiment the correct experiment is an immunofluorescence experiment.

Reviewer #2 (Comments to the Authors (Required)):

Overall, the manuscript and the proposed mechanism have improved in quality. The additional experiments have further substantiated the claims of the authors. In my opinion, the authors have addressed all comments sufficiently.

I suggest a change of the title as to my opinion there is residual doubt as to this TRAP1 variant causes autoinflammation.

Similarly, in the summary and throughout the manuscript, I would refrain the authors from making strong causal arguments about the effects of the discovered TRAP1 variant in the development of autoinflammation (which the authors indeed did in most instances).

Indeed, some findings implicating the found TRAP1 variant(s) as loss of function variants come from single patient studies (e.g. XPB-1 splicing, colocalization studies of TRAP1 protein). Also, a direct experiment, linking the TRAP1 variant with increased IL-18 secretion by immune cells is lacking.

As you will see, the reviewers appreciate the revision performed but think that a few remaining concerns need addressing. We would thus like to invite you to submit a final version of your manuscript, addressing the points still raised by the reviewers. Additionally, please also:

- indicate the corresponding author in the manuscript file right

Done – A Standing.

- add a COI statement

This has been added to the title page of the manuscript (No COIs)

Conflict of interest: The authors declare that the research was conducted in the absence of any commercial or financial relationships that could be construed as a potential conflict of interest.

- provide information on the patients' consent for publication of pictures

We confirm that the patients have consented to all aspects of the scientific work including publication of pictures presented in this manuscript. We have added this statement to the ethics section of our methods which now reads:

This study was approved by the Bloomsbury ethics committee (ethics number 08H071382); we obtained written informed consent from all the family members, patients, and healthy adult controls who participated. Anonymised paediatric age-matched control sera, surplus to routine clinical requirements, were obtained from the Immunology Laboratory at Great Ormond Street Hospital NHS Foundation Trust, and were used in some experiments without written consent, in accordance with ethical approval for the use of anonymised laboratory samples surplus to routine requirements. **All study subjects consented to publication of this report, including clinical images.**

We have also added blackout bars to figure 1A.

- There is a callout for Fig 1J in the manuscript text but no according panel in Fig1 - please fix

Thank you we have amended this typo.

- add scale bars to Fig3B and 4A

We have added these.

Reviewer #1 (Comments to the Authors (Required)):

General Remarks

"we in fact should have referred to the binding of infliximab to the complex formed by TNF (free and transmembrane bound) to TNFR1." This statement is also wrong. Infliximab binds to free TNF, it prevents TNF binding to the receptor, that's how it works, see DOI 10.1074/jbc.M112.433961. It does not bind to TNFR1 in any shape or form. I understand that the reference cited, Nedjai et al, hypothesised this happened but it does not.

We merely made a simple clinical observation regarding the autoinflammatory status which seemed a bit worse for the patient on infliximab, which reminded us of the story in TRAPS as suggested by Nedjai. It is clear this is still a contentious issue for this reviewer, however, and given that this is a relatively minor part of the story in our paper we have further simplified the sentence we have included, removing all mention of the mechanism which clearly remains speculative; the revised sentence is:

"Infliximab may provoke increased pro-inflammatory cytokine production in some patients with TRAPS (Nedjai et al., 2009); the mechanism of clinical deterioration in IV-1 is, however, unknown".

The manuscript still has a yawning logical disconnect. Sure TNFR1 mutations cause TRAPS accumulation, but isn't the point that the TRAPs mutation affects TNFR1 levels? Why is this not tested?

We are somewhat confused by the reviewer's terminology here- specifically "TNFR1 mutations cause TRAPS accumulation"- it is unclear what they mean by TRAPS accumulation: there is no such molecule as "traps"- TRAPS stands for TNF receptor associated periodic syndrome i.e. a description of the syndrome not the molecule. TRAP1 is a TNF receptor associated protein 1 that we describe in this paper, so we think that the reviewer is referring to TRAP1? We already show that TNFR1 mutations causing TRAPS result in intracellular co-localisation of TRAP1 with TNFR1 (figure 4A). We have described why that is logical if TRAP1 is chaperoning an abnormally folded intracellular TNFR1 protein molecule, as described in our manuscript. We expect mutated TRAP1 mutations cause a loss of function in this protein and do not expect an accumulation of TNFR1.

To me increased levels of TNFR1 in the serum of patients suggest that there is NOT a problem with TNFR1 production and presentation to the membrane, however, at best, this is a surrogate test. The direct test is to simply look at TNFR1 expression on the membrane using flow cytometry or IHC, as for example in Figure 4. If necessary the TRAPs mutants could be over expressed although patient 4 samples would be best.

We are making the point that TRAP1 associates with abnormally folded TNFR1 in patients with the syndrome TRAPS. Patients with TRAPS can have high low or normal circulating TNFR1 levels, despite having structural mutations in the receptor; so mutations in the receptor do not equate simply to circulating levels. The thinking currently is that it is not the amount of the receptor but rather how it is handled intracellularly that drives autoinflammation in TRAPS. We highlight the possible role of

TRAP1 in that process, deficiency of which can result in unfolded protein response and cellular stress, as demonstrated by our experiments and observations.

The Western blot looks encouraging however to control the specificity of the antibody in an immunofluorescence experiment the correct experiment is an immunofluorescence experiment.

Western blot is a recommended method to test antibody specificity for microscopy (Kurien et al., 2011). We demonstrate with a clear and specific staining pattern, with only a band of the correct size. Many other studies also have used Western blotting in this context. Also, as per our previous comments to the reviewers, we demonstrated that our TRAP1 siRNA knockdown cell lysate produced a reduced specific protein band (of the correct size) on Western blotting, so we are confident that our antibody is specific in these experiments. Since all human cells express TRAP1, you would need a complete knockout cell line to demonstrate absent staining, which we do not have so this is not so straightforward or clear cut. This specific antibody that we used was tested for immunofluorescence staining by the company during development ("Purified, Mouse, Anti-Hsp75,42/Hsp75, RUO - 612345 | BD Biosciences-Europe", 2019), and has been used for immunostaining in other published studies (Fismen et al., 2013; Pak et al., 2017).

References:

Fismen, S., D. Thiyagarajan, N. Seredkina, H. Nielsen, S. Jacobsen, T. Elung-Jensen, A.L. Kamper, S.D. Johansen, E.S. Mortensen, and O.P. Rekvig. 2013. Impact of the Tumor Necrosis Factor Receptor-Associated Protein 1 (Trap1) on Renal DNase1 Shutdown and on Progression of Murine and Human Lupus Nephritis. *The American Journal of Pathology* 182:688-700.

Kurien, B.T., Y. Dorri, S. Dillon, A. Dsouza, and R.H. Scofield. 2011. An Overview of Western Blotting for Determining Antibody Specificities for Immunohistochemistry. In *Signal Transduction Immunohistochemistry: Methods and Protocols*. A.E. Kalyuzhny, editor Humana Press, Totowa, NJ. 55-67.

Pak, M.G., H.J. Koh, and M.S. Roh. 2017. Clinicopathologic significance of TRAP1 expression in colorectal cancer: a large scale study of human colorectal adenocarcinoma tissues. *Diagnostic Pathology* 12:6.

Reviewer #2 (Comments to the Authors (Required)):

Overall, the manuscript and the proposed mechanism have improved in quality. The additional experiments have further substantiated the claims of the authors. In my opinion, the authors have addressed all comments sufficiently.

I suggest a change of the title as to my opinion there is residual doubt as to this TRAP1 variant causes autoinflammation.

Yes we agree and have amended to "TRAP1 chaperone protein mutations and autoinflammation", which addresses this reviewer's point here.

Similarly, in the summary and throughout the manuscript, I would refrain the authors from making strong causal arguments about the effects of the discovered TRAP1 variant in the development of autoinflammation (which the authors indeed did in most instances).

Indeed, some findings implicating the found TRAP1 variant(s) as loss of function variants come from single patient studies (e.g. XPB-1 splicing, colocalization studies of TRAP1 protein).

We have amended throughout as suggested by the reviewer, removing any strong causal arguments from the narrative.

Also, a direct experiment, linking the TRAP1 variant with increased IL-18 secretion by immune cells is lacking.

We have been able to generate some data to support this with the very last few remaining peripheral blood mononuclear cells which we had stored from patient 4. We found that the patient's PBMC in culture produced greater levels of both IL-18 and IL-1 than healthy control with stimulation, and also showed higher levels of caspase 1 activation, a prerequisite of IL-1 and 18 secretion, measured using the FLICA assay. We recognise at this point this data is from the only patient where cells are available (although importantly the only patient with only *TRAP1* and no *MEFV* mutation), so we hope that other investigators may be able to follow up this finding if more patients are identified in the future. We have incorporated the data into figure 5, whilst flagging this caveat.

December 20, 2019

RE: Life Science Alliance Manuscript #LSA-2019-00376-TRR

Dr. Ariane Shamilla Isaura Standing
UCL and Great Ormond Street Institute of Child Health
III
30 Guildford Street
London WC1N 1EH
United Kingdom

Dear Dr. Standing,

Thank you for submitting your Research Article entitled "TRAP1 chaperone protein mutations and autoinflammation". It is a pleasure to let you know that your manuscript is now accepted for publication in Life Science Alliance. Congratulations on this interesting work.

DISTRIBUTION OF MATERIALS:

Again, congratulations on a very nice paper. I hope you found the review process to be constructive and are pleased with how the manuscript was handled editorially. We look forward to future exciting submissions from your lab.

Sincerely,
